# The Welfare of Horses Competing in Three-Barrel Race Events Is Shown to Be Not Inhibited by Short Intervals between Starts

**DOI:** 10.3390/ani14040583

**Published:** 2024-02-09

**Authors:** Helio C. Manso Filho, Keity L. G. Trindade, Carolina J. F. L. Silva, Raissa K. S. Cruz, César F. Vilela, Clarisse S. Coelho, José D. Ribeiro Filho, Helena E. C. C. C. Manso

**Affiliations:** 1Núcleo de Pesquisa Equina, Universidade Federal Rural de Pernambuco (UFRPE), Recife 52171-900, PE, Brazil; keitylgtrindade@gmail.com (K.L.G.T.); carolinajonesfls@gmail.com (C.J.F.L.S.); helenaemiliamanso@ufrpe.br (H.E.C.C.C.M.); 2Faculdade de Medicina Veterinária, Centro Universitário Cesmac, Maceió 57051-160, AL, Brazil; raissa.cruz@cesmac.edu.br; 3Independent Researcher, Americana 13474-470, SP, Brazil; veterinarioperito@gmail.com; 4Mediterranean Institute for Agriculture, Environment and Development, Universidade de Évora, 7006-554 Évora, Portugal; clarisse.coelho@ulusofona.pt; 5Veterinary and Animal Research Centre (CECAV), Faculty of Veterinary Medicine, Lusofona University, 376 Campo Grande, 1749-024 Lisbon, Portugal; 6Departamento de Medicina Veterinária, Universidade Federal de Viçosa, Viçosa 36570-900, MG, Brazil; jdribeirofilho@gmail.com

**Keywords:** exercise, welfare, hormones, blood, cytokines, thermography, CK, fibrinogen, quarter horses

## Abstract

**Simple Summary:**

Welfare assessment of horses used in equine disciplines such as 3-barrels must incorporate both invasive and noninvasive methods. Ten Quarter Horses regularly participating in 3TB competitions were evaluated through infrared thermography and blood biomarkers to determine the effects of a short interval between races on their health and well-being. These animals performed two races of 3TB at 2 min intervals, and samples were obtained before and during the recovery period in four moments. The physiological responses observed in experimental horses indicated changes in some parameters within 1 h after races on surface temperatures but with no differences compared to pre-race records. Parameters usually used to assess well-being and health in athletic horses did not change. We conclude that well-conditioned 3TB horses are not harmed by running two races with short intervals between them.

**Abstract:**

Equestrian sports require precise animal welfare and health evaluations. To test the hypothesis that horses maintain their welfare when subjected to two three-barrel (3TB) races with 2 min intervals, an experiment was designed to evaluate their surface temperature using infrared thermography (IRT) in regions of interest (barrel, flank, neck, jaw, corner of the mouth, and ocular caruncle) and also measure blood biomarkers (hemogram, total plasma protein, fibrinogen, urea, creatinine, GGT, CK, cortisol, IL-6, and IL-1β). Ten Quarter Horses were monitored through thermography (pre-race, +1, +4, and +24 h post-race) and blood sampling (pre-race, +1, +4, and +24 h post-race). ANOVA and Tukey test at 5% were used. IRT in six regions of interest (Left/Right—barrel, flank, neck muscles) increased at +, with no differences between values recorded at +1 and +4 when compared to those measured pre-race (*p* > 0.05). Plasma protein, RBC count, hemoglobin, hematocrit, WBC count, neutrophils, and lymphocytes (*p* < 0.05) increased immediately after the races, with recovery at +1 h. Other biomarkers did not change, including cortisol, IL-6, and IL-1β (*p* > 0.05). Results indicate that well-conditioned 3TB horses subjected to two races at short intervals do not show changes that could be related to impaired health or welfare.

## 1. Introduction

Different invasive and noninvasive methods have been used to evaluate the welfare of athletic horses from numerous disciplines. Findings also considered the level of experience of the horses and the riders and the pre-competition conditions of the horses (training, rest periods, diet, etc.), including the numerous causes of poor performance and gut microbiota composition [1,2,3,4,5,6,7,8,9,10]. These different studies expand general knowledge about the impacts of physical exercises that occur in training and competitions and can be used in the evaluation of the welfare of horses used in different equestrian disciplines, whether classical (racing, jumping, etc.) or contemporaneous (three-barrel racing, *vaquejada*, *paleteadas*, etc.). Particularly in the latter, information is useful as there is a range of participants of different breeds in addition to the most varied levels of knowledge about the training techniques. Some studies have indicated that less experienced horses are more likely to have welfare impairment during competitions and training [6,8,10].

The three-barrel event (3TB), which consists of the rider and the horse running around three barrels distributed in a triangular shape along the course in the shortest time possible [11,12], is a popular equestrian event in the Americas. In these tests, the animals move at speeds above 25 km/h for less than 20 s, and there may be repetitions of the race after a short time interval. Due to these characteristics, authors have investigated the influence of this type of physical effort, both for the evaluation of performance and for the possible diseases resulting from 3TB races [11,13,14,15,16,17,18]. However, few articles have associated health and welfare assessment methods in 3TB horses, such assessments being more common in other equestrian modalities, such as *vaquejada* [2,19], running [20,21,22], endurance [4,23], and jumping exercises [10,24,25].

To test the hypothesis that horses subjected to two 3TB races at a short interval period may experience some degree of welfare impairment, this novelty study evaluated Quarter Horses subjected to two races 2 min apart through noninvasive methods (infrared thermography, IRT), and invasive methods (blood sampling for analysis of complete blood count, plasma proteins, fibrinogen, urea, creatinine, γ-glutamyl transferase (GGT), creatine kinase (CK), cortisol, interleukin-6 (IL-6), interleukin-1β (IL-1β)). We expect changes related to the level of the imposed effort but without interfering with the quality of life of the used animals.

## 2. Materials and Methods

### 2.1. Animals and Breeding System

Ten Quarter Horses (age: 8–10 years; average weight: 480 kg; 5 geldings and 5 mares) were housed in a training center for this equestrian sport in Recife, Pernambuco, Brazil (−7.94540; −34.91026). These 3TB horses were housed in boxes with ample natural ventilation, no direct sunlight, and a minimum ceiling height of 2.5 m and a maximum height of 4.0 m. These boxes provided a minimum area of 16 m^2^/horse.

They were fed fresh and chopped elephant grass (*Pennisetum purpureum* Schum.), approximately 15.0 kg/day, divided into three meals, and Tifton grass hay (*Cynodon* spp.) in slow-feeder baskets ad libitum overnight. They also received supplementation with commercial concentrate (Equitage with laminates, CP: 12% (min); EE: 9% (min); GF: 10% (max); MM 10% (max); digestible energy 3.9 MCal/kg (min), Guabi Nutrição Animal, Campinas, São Paulo, Brazil) and were supplemented at 5.0 kg/animal, divided into 3 meals of equal weight daily. Mineralized salt and water were available ad libitum. The body mass of the horses was determined using an equine weighing tape.

All animals used in this study had competed regularly in the last three months and were considered healthy in previous clinical and orthopedic examinations. They were trained three times a week for 30/40 min per day. Their exercise was made up of walking, trotting, cantering, and specific training sessions with running between barrels. In addition, twice a week, they were walked for 60 to 90 min.

### 2.2. Three-Barrel Race Test

The 3TB races were performed between 7:00 and 9:00 A.M. on a soft sand track, according to the official descriptions [26]. The barrels were arranged on the track in the form of a triangle, with 27.5 m between the first and second and 32 m from the third to the other barrels. The distance between the first and second barrels from departure/arrival was 18.30 m, according to official standards [26]. The horses were subjected to a 10 min warm-up, with short trotting and gallops. Then, they ran twice across the track with a 2 min interval between races. After the 2nd race, they underwent a recovery period of approximately 10 min with the horses at the walk and mounted on the track, adapting the test model previously described.

The evaluation of the animals started 24 h before the races and continued for 24 h after them, applying the evaluation model of the 5 domains [27], the standards of the Association of Quarter Horse Breeders (ABQM) [26], and Brazil’s Ministry of Agriculture, Livestock, and Food Supply (MAPA).

### 2.3. Collection of Blood Samples

All blood samples were obtained after a fast of at least 10 h by jugular vein puncture in tubes with EDTA for complete blood count and in heparinized tubes for biochemical analysis. These latter samples were centrifuged after obtaining the plasma, which was stored in a −20° freezer until the material was processed. These samples were collected in the following moments: pre-race/fasting, immediately after the second 3TB race, and +1, +4, and +24 h after the 2nd race.

The hematological analyses were performed in semiautomatic equipment (Roche Poch 100iv, São Paulo, SP, Brazil) to obtain red blood cell count, hemoglobin concentration, hematocrit, mean corpuscular volume (MCV), mean corpuscular hemoglobin concentration (MCHC), red blood cell distribution width as its standard deviation (RDW-SD), red blood cell distribution width as its coefficient of variation (RDW-CV), total leukocyte count, differential leukocyte count, platelet count, and other white cell count. Plasma proteins were determined by refractometry, and fibrinogen was determined by heat precipitation (56 °C) and refractometry.

The plasma CK, GGT, urea, and creatinine were measured in semiautomatic equipment (Doles D-250, Goiânia, Brazil) using commercial kits (Doles, Brazil) following the manufacturer’s recommendations. The analyses of cortisol, IL-6, and IL-1β were performed by enzyme-linked immunosorbent assay (ELISA) in semiautomatic equipment (Bioclin Mindray MR-96A, 837, Belo Horizonte, Brazil) using commercial kits (Human Cortisol ELISA, Elabscience, E-EL-0157; Human IL-6 ELISA, Elabscience, E-EL-H6156; Human IL-1β ELISA, Elabscience, E-EL-H0149, Houston, TX, USA).

### 2.4. Thermography

An infrared thermograph (E4 Wifi, FLIR Systems AB, Tabi, Sweden) was used in the following phases: pre-race, with the animals fasted, and at +1, +4, and +24 h after the 2nd race of the 3TB. They were performed in an indoor environment, without air current and with the animals dry at an average distance of 150 cm from the region of interest. As described above, after the races, all animals underwent a recovery period of approximately 10 min with the horses at a walk and then were sent to their boxes, with available water and fresh forage, without being bathed so that the body dried naturally to not compromise the thermographic images. The regions evaluated were the ocular caruncle, buccal commissures (corner of the mouth), masseter (*Masseter*) muscles, neck with emphasis on the area of the trapezius (*Trapezius Cervicus*) and brachiocephalic (*Brachiocephalicus*) muscles, the barrels/costal arches, and the flanks, all on both sides except the caruncle only on the left, as described and adapted from Silva et al. [2] (Table 1). Left and right measurements were analyzed separately.

The emissivity for the readings was set at 0.95 ε in a previously described protocol [28,29,30,31]. These images were evaluated in a software application (FLIR Tools+ 6.4.18039.1003, FLIR Systems AB, Wilsonville, OR, USA) to adjust them according to the temperature and relative humidity of the environment at the time of collection. The ambient temperature and relative humidity were measured by a dithermohygrometer (Thermo-Hygrometer #7666.02.0.00, Incoterm Indústria de Termómetros LTDA, Osório, Brazil) with a temperature resolution of 0.1 °C and accuracy of +1 °C to 0 °C. At 50 °C, the relative humidity had a resolution of 1% (precision + 5%), according to the manufacturer’s instructions.

The training and housing environmental conditions and thermal risk rate were evaluated using the thermal stress index (TSI) and the temperature/humidity index (THI) [32,33,34,35]. The formula for the TSI was the sum of the ambient temperature in Fahrenheit (°F conversion = (5/6 × °C) + 32) and the relative humidity. The following formula was used to calculate the THI: 0.8 × T (°C) + RH × (T − 14.4) + 46.4, with moisture expressed numerically (75% equals 0.75) (Table 2).

### 2.5. Statistical Analysis

The results were analyzed through one-way ANOVA and Tukey’s test to determine differences between the experimental moments in each region of interest, but without comparing the left and right sides. In all cases, the significance level was set at 5%, and the results are expressed as mean ± standard error. The statistical analysis software SigmaPlot 13.0 (Systat Software Inc., San Jose, CA, USA) was used in all analyses.

## 3. Results

According to the hematobiochemical analyses, there were variations in the counts of red blood cells, white blood cells, neutrophils, lymphocytes, platelets, hemoglobin, hematocrit, and total plasma protein levels (*p* < 0.05) (Table 3). These parameters reached higher values immediately after the 3TB races, with significant decreases as early as +1 h, except for the platelet count, which remained high even after 4 h.

Thermographic images of 13 regions of interest were evaluated in this study. Results showed significant variations in six of these regions (left/right—barrel, flank, neck muscles), with the highest values recorded at +1 h and the lowest at +24 h (*p* < 0.05) (Table 4). Environmental conditions at the training center were also evaluated and indicated typical values for the Atlantic Forest in the State of Pernambuco. Temperatures ranged from 24 to 29 °C, and relative humidity ranged from 71 to 79%. These values indicate level 3 and mild TSI and THI, respectively, in the 3TB horses while fasted (pre-race) and 1 h post-race. However, at +4 h and +24 h, the THI ranged from mild to moderate.

## 4. Discussion

The study confirmed the hypothesis that 3TB horses would not suffer any harm by participating in two races shortly after each other, following the current rules. The study found that horses can adapt to effort under current climatic conditions and official regulations.

In the current study, it was observed that the hematocrit levels increased by approximately 56%, along with a 15% increase in plasma proteins. This indicates the impact of physical effort on these parameters, which is consistent with previous studies conducted on animals in the same equestrian modality [12,17]. The increase in the concentration of red blood cells, accompanied by the increase in hematocrit, is due to the contraction of the spleen, favoring the transport of oxygen to the tissues. This process is accompanied by a temporary hemoconcentration, as with the recovery of the horse after exercise, the concentrations of red blood cells and hematocrit return to the values observed in the horse at rest. These processes are more efficient in well-conditioned horses [12,19], such as those used in the current research.

Additionally, the levels of leukocytes, lymphocytes, and platelets increased due to the release of epinephrine, which is a common occurrence during such physical activities and returned to values like those observed in the pre-race period. The changes observed in white blood cells were like previous studies [12,17], where there is an increase in the concentrations of these white cells followed by normalization. Souza et al. [12] observed that there may be a reduction in the concentration of lymphocytes and that this process may be related to the level of intensity and duration of exercise and the level of conditioning of the horses evaluated. Reduction in the concentration of lymphocytes may compromise the immunity of horses during the recovery period and should be further analyzed in new trials [12]. In the current test, this reduction was not identified, but the concentrations in the period +4 h are numerically lower than that observed in the pre-race. 

Blood biomarkers such as fibrinogen, GGT, and CK are used to evaluate athlete horses during training and after races. Fibrinogen is a protein that increases after intense exercise, indicating inflammation in body tissues. If fibrinogen increases by more than 25%, it suggests moderate inflammation [36]. GGT, which is used as a biomarker of fatigue, was lower than 50 IU/L in all measurements. Concentrations above 50 IU/L in racehorses suggest overtraining or fatigue, and there is also a genetic component to its increase [4,37]. The study used quarter horses, which include racehorses, so GGT should be measured in 3TB athletes. CK concentrations indicate muscle damage when above 400 IU/L after 4–6 h of completing exercise, but they may be elevated immediately after exercise due to hemoconcentration [12,19,38]. However, in the horses tested here, fibrinogen concentrations did not change, and CK increased to 280 IU/L within +4 h but remained below 400 IU/L. Furthermore, no significant variations in blood urea and creatinine concentrations were observed, indicating that there was no more post-race proteolysis than expected [17].

During a typical 3TB race, there is usually a rise in hematocrit and total plasma protein levels, along with health biomarkers, such as fibrinogen and CK. However, it has been observed that there are no increases in cortisol, IL-6, or IL-1β because of physical exertion. This is consistent with previous research on quarter horses that participated in *vaquejada* [2,19], a sport like 3TB running in terms of its high intensity and short duration.

Acute stress can lead to an increase in various blood biomarkers, which act as protectors against changes in homeostasis during exercise [7]. However, good physical conditioning can reduce the stress load and promote adaptive metabolic processes associated with exercise. This can also be reflected in the absence of variation in these biomarkers [7,18,39,40]. Studies have shown that cortisol, IL-6, and IL-1β remain stable during training, leading to improved performance in different equestrian sports, including classical and fieldwork on farms [4,12,20,39,41]. Similar findings were observed in the present experiment, and this absence of changes in the concentrations of cortisol, IL-1b, and IL-6, which could impact different aspects of the animals’ immune system, when combined with the results of variations in the concentrations of leukocytes and lymphocytes, corroborates the statements of these authors. 

Several authors have emphasized the significance of the correlation between different blood biomarkers and the surface temperatures obtained through infrared thermography (IRT). This combined assessment can help in observing post-race inflammatory processes [31], and aid in comprehending the adaptations to different types of exercise. Furthermore, sharing the results with the public can have a positive impact by demonstrating the metabolic adaptations that occur during exercise [42].

During the experiment, significant variations in surface temperatures were observed in the lateral areas of the horses, including their barrels, flanks, and brachycephalic muscles. It is unclear why this happened because different factors can affect the horse’s surface temperature before, during, and after exercise [30,43]. However, it is worth noting that these areas are more prone to heat dissipation, especially when the horses are at rest or performing exercise in hot and humid weather [44]. During the current experiment, the environmental temperature was above 25 °C, and the relative humidity was above 70%, which may have contributed to the observed results.

The 3TB horses in the experiment performed short-duration, high-intensity exercises, which led to greater heat dissipation during their post-race recovery. This dissipation is essential to maintain the animal’s homeostasis and overall health and welfare [44,45]. Poor performance and physical conditioning may be due to the loss or reduction of thermoregulatory capacity, which can damage animal welfare [45]. Therefore, it is crucial to establish effective cooling methods during training and competitions, especially in tropical regions where high temperatures and humidity are common [46].

During high-speed sports, the body generates an oxygen debt, which is corrected during the post-race phase through tachypnea. This process not only helps to replenish oxygen levels but also contributes to heat loss [34,45]. Additionally, muscle contraction may also affect both systemic and local temperatures [21], particularly in regions that experience greater muscle activity due to turns around the barrels and tachypnea. Therefore, these regulatory mechanisms could be potential factors that cause the elevation of IRT temperature in certain areas of interest.

There was no change in temperature in the corners of the mouth and masseters 24 h after the runs. Changes in temperature in these regions after some hours of the exercise may indicate inflammation. These areas are affected by the bits and harnesses used to control the horse while the rider is riding it. However, there were no other evaluations in these areas of interest besides *vaquejada* horses, which limits our understanding of the results in 3TB horses. The collection of IRT images 1 h after the races, the number of races, and the use of bridles and accessories may have affected the results. Nevertheless, the results are important because they indicate that there is no inflammation caused by the bits and harnesses commonly used in this equestrian sport, provided they comply with the official rules. Similar results were observed in these areas in *vaquejada* horses [2], where no changes in local temperature were observed.

The temperature of the eye caruncle, measured by infrared thermography, is considered a reliable indicator of core temperature and welfare in different species [30,43,44,47]. However, the present study found no significant variation in the temperature of the eye caruncle, which might be related to the measurement being taken only 1 h after physical exertion. Nonetheless, horses that participated in *vaquejada* races, which are intense and short-duration exercises, showed a significant increase in the temperature of the eye caruncle under climatic conditions like those of the present study.

It is important to note that the temperature in the eye caruncle may be influenced by the regulatory mechanisms of the hypothalamus–pituitary–adrenal axis [22,25,47], which could indicate a loss of well-being and chronic stress if the temperature remains elevated for a prolonged period [7,11,25,30,44]. In contrast, a lack of, or a temporary change in, local temperature due to physical effort has been associated with good physical conditioning [25].

As we have discussed before, horses have individual thermoregulatory responses that are closely related to the type of physical effort they perform. It is important to understand these responses to work with the animals while maintaining their well-being [45,48]. In this study, we evaluated the impact of environmental conditions using the TSI and THI, and our findings indicate that sweating plays a crucial role in thermoregulation for horses after physical exertion, especially in conditions with high-risk rates for TSI and THI [33,34,35]. The horses in our study were able to regulate their body temperature effectively through sweating, thanks to their short coat and their prior adaptation to the environmental conditions of the evaluation sites.

The surface thermographic parameters and blood biomarker concentrations can be affected by climatic conditions. In this experiment, the highest average temperature of approximately 29.3 °C was observed four hours after physical activity when the THI was 80.4. During that time, the temperatures recorded by IRT related to the flanks, costal arches, and brachycephalic muscle had not yet normalized, and neither had the protein levels, indicating some degree of hemoconcentration. However, the horses did not exhibit any clinical signs of dehydration, and their hematocrit returned to the pre-race values. At this stage, the horses had free access to water, forage, and salt, so these findings were not expected. Further studies with 3TB horses are required to evaluate plasma volume and the effect of feeding on these parameters.

We conducted a trial that had several factors that may have affected the outcomes. A small sample size was used because we focused on a group of well-trained animals that were all undergoing the same training program, which included track and feeding management. Another important factor to consider is the local climatic conditions, which were quite specific. The average temperature was above 25 °C, and humidity was always over 60%. These conditions are common in the Americas during certain times of the year, and they can increase the difficulty for athlete animals in tropical regions and harm their health and well-being. Lastly, we did not take any images using IRT immediately after the races due to high sweating during this phase. IRT images can be affected by surface moisture, which means that high sweating levels could indicate temperatures that are above or below the expected temperature due to the accumulation of sweat on the surfaces of interest.

After the races, some biomarkers, such as red blood cells, hemoglobin, hematocrit, leukocytes, lymphocytes, and plasma proteins, increased in the horses. These are typical physiological responses for animals that must perform short-duration and high-intensity exercise. However, the levels of these biomarkers returned to pre-exercise values within the first hour of recovery. Other blood parameters that indicate the welfare and health of athlete horses, such as eye caruncle temperature, CK, GGT, fibrinogen, cortisol, IL-6, and IL-1β, remained unchanged. Thermographic evaluations showed a transient increase in surface temperature at only six points of interest 1 h after the races. However, there were no clinical changes.

Finally, analyzing the results presented here, it can be suggested that the combination of the current evaluation model and the inclusion of other biomarkers of well-being could indicate the path to better post-exercise physical recovery, including the relationships between interleukins, immunological aspects, and adaptation to exercise, which are still little known in non-traditional sports, improving the health and well-being of these horses.

## 5. Conclusions

After the 3TB races, horses showed typical physiological responses for animals that must perform short-duration and high-intensity exercise. However, the levels of these biomarkers returned to pre-exercise values within the first hour of recovery. Other blood parameters associated with the welfare and health of athlete horses, such as eye caruncle temperature, CK, GGT, fibrinogen, cortisol, IL-6, and IL-1β, remained unchanged. Thermographic evaluations showed a transient increase in surface temperature at only six points of interest 1 h after the races. However, there were no clinical changes. Therefore, we conclude that well-conditioned horses of the 3TB sport are not harmed by running two races shortly after one another.

## Figures and Tables

**Table 1 animals-14-00583-t001:** Regions of interest on the horses for obtaining thermographic images. The red dot indicates the central point of the area of interest where the temperature was taken.

Region and Description	Location	Region and Description	Location
Ocular caruncle: region at the inner angle of the eye	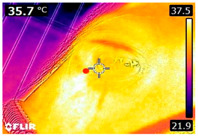	Jaw/Masseter muscle region: Region on the lateral surfaces of the head, with the masseter muscle	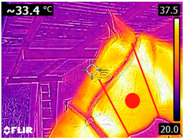
The corner of the mouth: the point where the lips meet at the corner of the mouth	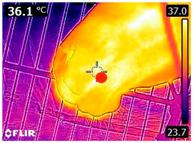	Neck/Trapezius muscle region: Muscle near the upper border of the neck, near the base (1: upper edge of neck; 2: withers; 3: midpoint of base of neck)	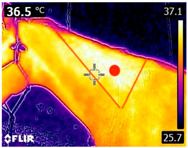
Neck/Brachiocephalicus muscle region: muscle near the lower border of the neck and dorsal to the jugular groove	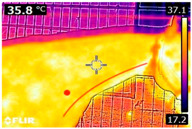	Barrel/Costal arches: region referring to the costal arches (1: elbow; 2: stifle; 3: point of hip)	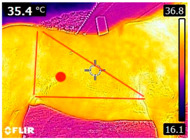
Flank: midpoint between the point of the hip and the stifle	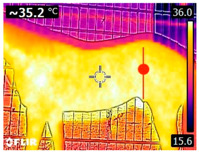		

**Table 2 animals-14-00583-t002:** Thermal stress index (TSI) and temperature and humidity index (THI) risk ratings.

Model	Index	Risk Rating	Effects on the Metabolism of Horses
TSI	1	<120/130	Normal cooling of the horse’s body through evaporation, respiration, and sweating unless the horse is obese or very hairy.
	2	>140	Sweat is responsible for heat loss, which can be aggravated by obesity or excessive hair.
	3	>150	Evaporative heat loss is compromised, especially if the relative humidity is above 50%, but sweat is still important for heat loss.
	4	>180	The natural dissipation of heat does not occur, which may raise body temperature. There is great danger of thermal stress.
THI	Light	72–79	Mild risk of thermal stress. Heat loss is done by sweating. Hairy or obese horses may be more impacted.
	Moderate	80–89	Moderate risk of heat stress.
	Severe	>90	Severe risk of heat stress.

**Table 3 animals-14-00583-t003:** Results of the blood tests for the test horses of the three barrels subjected to two races with a 2 min interval.

Biomarkers	Experimental Periods				
	Pre-Race	Immediately after the Races	+1 h after the Races	+4 h after the Races	+24 h after the Races
Red blood cell count, ×10^6^/µL	6.85 ± 0.30 ^B^	10.79 ± 0.36 ^A^	7.44 ± 0.29 ^B^	7.48 ± 0.26 ^B^	7.18 ± 0.41 ^B^
Hemoglobin, g/dL	11.16 ± 0.51 ^B^	17.57 ± 0.57 ^A^	12.06 ± 0.45 ^B^	12.12 ± 0.40 ^B^	11.60 ± 0.60 ^B^
Hematocrit, %	31.27 ± 1.33 ^B^	49.97 ± 1.68 ^A^	33.74 ± 1.29 ^B^	33.76 ± 1.14 ^B^	33.43 ± 1.80 ^B^
MCHC, g/dL	35.66 ± 0.27	35.18 ± 0.19	35.52 ± 0.25	35.95 ± 0.21	35.84 ± 0.24
MCV, fL	45.29 ± 0.49	46.36 ± 0.49	45.62 ± 0.49	45.20 ± 0.50	45.18 ± 0.50
RDW-CV, %	19.53 ± 0.30	19.63 ± 0.30	19.65 ± 0.31	19.71 ± 0.31	19.46 ± 0.27
RDW-SD, fL	35.61 ± 0.44	36.44 ± 0.51	35.92 ± 0.46	35.65 ± 0.48	35.52 ± 0.45
Platelets, ×10^3^/µL	117.40 ± 7.80 ^A^	143.00 ± 11.90 ^AB^	145.20 ± 9.13 ^AB^	160.90 ± 6.94 ^A^	115.80 ± 1.74 ^B^
White blood cells, ×10^3^/µL	7.00 ± 0.49 ^C^	9.54 ± 0.44 ^A^	7.72 ± 0.42 ^BC^	9.26 ± 0.40 ^AB^	7.41 ± 0.20 ^C^
Lymphocytes, ×10^3^/µL	2.26 ± 0.17 ^B^	3.79 ± 0.22 ^A^	2.45 ± 0.14 ^B^	2.08 ± 0.12 ^B^	2.29 ± 0.16 ^B^
Other white blood cells, ×10^3^/µL	4.74 ± 0.48 ^B^	5.75 ± 0.48 ^AB^	5.27 ± 0.48 ^B^	7.18 ± 0.46 ^A^	5.12 ± 0.30 ^B^
Urea, mg/dL	54.54 ± 3.98	58.75 ± 4.32	58.04 ± 3.47	63.38 ± 2.18	53.36 ± 2.84
Creatinine, mg/dL	2.29 ± 0.24	2.61 ± 0.28	2.69 ± 0.29	2.43 ± 0.24	2.20 ± 0.22
Plasma proteins, mg/dL	6.52 ± 0.17 ^B^	7.46 ± 0.13 ^A^	6.52 ± 0.17 ^B^	6.88 ± 0.15 ^AB^	6.58 ± 0.14 ^B^
Fibrinogen, mg/dL	0.32 ± 0.08	0.38 ± 0.06	0.40 ± 0.04	0.56 ± 0.10	0.41 ± 0.06
GGT, IU/L	45.72 ± 2.55	45.07 ± 2.56	46.60 ± 2.62	42.53 ± 3.02	45.00 ± 1.87
CK, IU/L	208.16 ± 31.44	236.38 ± 21.42	231.10 ± 25.74	282.10 ± 24.42	213.58 ± 14.03
Cortisol, ng/dL	27.69 ± 7.57	26.13 ± 4.84	24.75 ± 5^.^69	23.40 ± 5.36	23.27 ± 6.52
IL-1β, pg/dL	39.81 ± 2.24	35.88 ± 2.29	39.10 ± 3.11	38.38 ± 3.52	37.99 ± 3.63
IL-6, pg/dL	3.96 ± 0.19	3.54 ± 0.25	4.25 ± 0.43	3.71 ± 0.43	4.15 ± 0.19

Notes: Different uppercase letters in the same row denote significant differences according to Tukey’s test (*p* < 0.05). GGT; γ-glutamyl transferase; CK: creatine kinase.

**Table 4 animals-14-00583-t004:** Ambient temperature, relative air humidity, thermal stress index, temperature/humidity index, and analysis of thermographic images for the test horses subjected to two 3TB races at a 2 min interval.

Parameters	Experimental Periods			
	Pre-Race	+1 h after the Races	+4 h after the Races	+24 h after the Races
Ambient temperature, °C (°F)	28.0 (82.4)	28.0 (82.4)	29.3 (84.7)	26.0 (78.8)
Relative humidity, %	79.4	72.0	71.5	77.6
IST	161.4	154.0	156.2	156.4
THI	79.6	78.6	80.4	82.0
Barrel/Costal arches, L	35.65 ± 0.39 ^AB^	36.68 ± 0.45 ^A^	36.10 ± 0.25 ^AB^	34.75 ± 0.34 ^B^
Barrel/Costal arches, R	35.43 ± 0.40 ^AB^	36.52 ± 0.51 ^A^	35.83 ± 0.25 ^AB^	34.90 ± 0.19 ^B^
Flank, L	35.83 ± 0.42 ^AB^	36.98 ± 0.34 ^A^	36.26 ± 0.21 ^AB^	35.04 ± 0.32 ^B^
Flank, R	35.73 ± 0.49 ^AB^	36.85 ± 0.40 ^A^	35.93 ± 0.21 ^AB^	35.26 ± 0.31 ^B^
Neck/Brachiocephalicus region, L	35.35 ± 0.43 ^AB^	36.36 ± 0.57 ^A^	35.64 ± 0.31 ^AB^	34.40 ± 0.37 ^B^
Neck/Brachiocephalicus muscle, R	35.47 ± 0.41 ^AB^	36.43 ± 0.37 ^A^	35.49 ± 0.28 ^AB^	34.55 ± 0.37 ^B^
Neck/Trapezius muscle, L	34.82 ± 0.85	35.29 ± 0.90	35.07 ± 0.55	33.65 ± 1.10
Neck/Trapezius muscle, R	35.66 ± 0.52	36.07 ± 0.41	35.06 ± 0.62	34.90 ± 0.45
Jaw/Masseter muscle region, L	35.48 ± 0.45	35.79 ± 0.36	35.77 ± 0.24	35.25 ± 0.24
Jaw/Masseter muscle region, R	35.39 ± 0.44	35.62 ± 0.40	35.71 ± 0.51	35.13 ± 0.30
The corner of the mouth, L	36.54 ± 0.27	36.40 ± 0.36	36.77 ± 0.28	36.00 ± 0.20
The corner of the mouth, R	36.52 ± 0.31	36.45 ± 0.28	36.71 ± 0.25	35.89 ± 0.26
Ocular caruncle, L	36.46 ± 0.60	36.10 ± 0.31	36.06 ± 0.32	35.60 ± 0.34

Notes: Different uppercase letters in the same row denote significant differences according to Tukey’s test (*p* < 0.05). L: left; R: right.

## Data Availability

The data presented in this study are available on request from the corresponding author.

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
