# Peer review of "The Welfare of Horses Competing in Three-Barrel Race Events Is Shown to Be Not Inhibited by Short Intervals between Starts"

_animals, 2024, doi:10.3390/ani14040583_

Round 1
Reviewer 1 Report
Comments and Suggestions for Authors
Review
Short intervals between races do not interfere with the welfare of three-barrel horses
REVIEWER: OVERALL IMPRESSIONS – this manuscript bares significant merit to the industry of barrel racing. The references to management/welfare are noted and applicable. In general, the grammar is good and the content well organised. Any concerns are listed in corresponding section below.
TITLE
The title is vague – suggest a title explaining the experiment further – for example - “The welfare of horses competing in three-barrel race events are shown to be not inhibited by short intervals between starts.”
SIMPLE SUMMARY
The simple summary should be easy for the layperson to read and in fact this summary is not. In its present form it reads more as an abstract, which is where it is better suited. Too many acronyms without the full wording (all acronyms in the simple summary and abstract must be fully worded in both sections, even though some will be repeated e.g. CK).
Line 28. ‘an experiment was prepared’ – suggest ‘an experiment was designed’.
Line 33. Suggest (Left/Right – barrel, flank, neck muscles)
ABSTRACT
Comments above.
Line 41. At the start of the sentence ‘10’ should be spelt out to read ‘Ten’.
KEYWORDS
Keys words need to help capture searches – suggest including – Quarter Horse; three barrel race; The acronym CK needs to be spelt out.
INTRODUCTION
Good introduction but would revise the first sentence into 2 sentences. Suggest –
Different invasive and noninvasive methods have been used to evaluate the welfare of athletic horses from numerous disciplines. Findings also considered the level of experience for each horse, rider, and the pre-competition conditions (training, rest periods, diet, etc.), including the numerous causes of poor performance and gut microbiota composition [1,2,3,4,5,6,7,8,9,10].
The authors need to elaborate on the barrel configuration further – as there are three barrels, the horse and rider complete the pattern with two left turns and one right turn. This pattern creates a disproportionate level of muscle usage during the event.
MATERIALS AND METHODS
Well written and easily repeatable.
Need to include the number of males (separate geldings) and females.
As per the introduction – the barrel test needs to be explained further as this might influence the test results.
The authors need to spell out all acronyms for the first time before applying them throughout the manuscript.
Line 147: When referencing the muscles, be specific and mention these are superficial muscles and name them correctly in italics e.g. trapezius cervicus. This includes Figure 1, which looks more like a table than figure.
Figure 2 should be Table 1, and all others corrected numerically thereafter, including references in the text.
RESULTS
Change the table numbers are per the recommendation from Material and Methods.
Line 198. Suggest (Left/Right – barrel, flank, neck muscles)
DISCUSSIONS
Paragraph beginning line 257: Consider the asymmetrical load through the configuration of turns, as this is not addressed or explained in the experiment.
I see no mention of variables between males and females because this was not addressed in M and M. However, if gender variables are present then it would have been interesting to note any changes associative to oestrus, for example, as per its relevance in Thoroughbred racing.
CONCLUSION
Relevant to the study.
REFERENCING
References 44 and 45 out of sequence line 260-262.
Comments on the Quality of English Language
In general, the quality of the English language is good.
Author Response
I attached one file with our comments and corrections.

Reviewer 2 Report
Comments and Suggestions for Authors
The manuscript entitled " Short intervals between races do not interfere with the welfare of three-barrel horses” is interesting. Scientifically sound studies investigating the importance of welfare assessment of horses used in equine disciplines such as 3-barrels must incorporate both invasive and non-invasive methods are rare. Thus, the approach of the study appears original. The contents of the manuscript are quite interesting because of his methodology and the tools of quantification used. I find it interesting. I thus find that this paper definitively delivers results that will surely be of interest to the readership of the journal Animals.
However, some corrections are needed. There are numerous corrections in the syntax and language of the abstract and throughout the paper.
– The title accurately conveys the essence of the study, which is commendable. However, for more specificity, consider: “Short intervals between races do not interfere with the welfare of three-barrel horses based on immunological and infrared thermography evaluation”
– The abstract is comprehensive but might be more concise by deleting the nonsignificant findings. This would streamline the flow for readers. Additionally adding the p-value is necessary, especially in the abstract. The simple summary may be changed with the abstract.
– In the introduction, when authors are characterized by different techniques for the welfare of horses used in different equestrian disciplines is unprecise. It should be mentioned that different markers may be evaluated such as cortisol, the anabolic index (testosterone to cortisol ratio) which was evaluated in race and endurance horses as well as even leisure horses in which intense exploitation did not influence the hormonal reaction. It might be helpful to briefly touch upon why this is of particular importance in the context of equine health, giving the reader a clearer picture of the study’s significance.
– The segment discussing the consequences of exercise on immune reaction should be added. The importance of this part should be highlighted. In my opinion, if authors measured some cytokines, it should be mentioned why they are important. There are several studies about the usage of different cytokines such as MMP2, IGF1, IL-13, and IL-1ra as markers of fitness status in horses as well as about novel inflammatory cytokines in different types of exercise.
– The objective statement at the end is clear, but it might benefit from an added emphasis on its novelty or what gap in the literature this study aims to address. This could further highlight the study’s significance.
– In the methods, specify the location or facility where the study took place as well as environmental conditions. Add the body condition score value (BCS).
- How fast was the serum centrifuged after collection?
- Why did authors not use the equine tests? Are those mentioned validated for horses? Human Cortisol ELISA, Elabscience, #E- 136 EL-0157; Human IL-6 ELISA, Elabscience, #E-EL-H6156; Human IL-1β ELISA, Elabscience, 137 #E-EL-H0149
– The discussion paragraph on hematological changes seems quite poor. Breaking it into smaller paragraphs, each focusing on a distinct point or aspect of RBC changes, WBC changes, and neutrophil/lymphocyte ratio, might make it more digestible for readers. The peripheral blood mononuclear cells proliferation, phenotype, functions, and reactive oxygen species production changes during exercise in horses as well as β2-Adrenergic Receptor regulation which is stimulated by exercise. Additionally, to reduce immune cell reaction some supplements are used such as green coffee extract which is currently of great interest to researchers due to its high concentration of chlorogenic acid (CGA) and its potential health benefits.
– In the section about CK. This can be potentially confusing for readers. Muscle fatigue may also be confirmed based on hemogram values and CPK and AST activity. After the strenuous exercise, CPK activity increases 4–35-fold whereas AST activity increases 2–6-fold. Try to clarify the distinction between CK increase after fatigue and muscle injury.
– Throughout the discussion, references to previous studies and your own study’s results are intermixed. Consider reorganizing for clearer flow, possibly by presenting established knowledge first, followed by how your findings align or differ.
- The last part of the discussion might benefit from a mention of any limitations in the study or areas where further research is recommended. This can provide context and direction for future studies in this area.
– The conclusion succinctly recaps the main findings. However, it might be helpful to add a sentence or two about the implications of these findings for future research or practical applications.
Comments on the Quality of English Language
Minor corrections are needed.
Author Response
I attached one file with comments and corrections.
